# Experimental Investigation on the Corrosion Detectability of A36 Low Carbon Steel by the Method of Phased Array Corrosion Mapping

**DOI:** 10.3390/ma16155297

**Published:** 2023-07-27

**Authors:** Jan Lean Tai, Rafał Grzejda, Mohamed Thariq Hameed Sultan, Andrzej Łukaszewicz, Farah Syazwani Shahar, Wojciech Tarasiuk, Arkadiusz Rychlik

**Affiliations:** 1Department of Aerospace Engineering, Faculty of Engineering, University Putra Malaysia, Serdang 43400, Malaysia; taijanlean2008@hotmail.com (J.L.T.); farahsyazwani@upm.edu.my (F.S.S.); 2Faculty of Mechanical Engineering and Mechatronics, West Pomeranian University of Technology in Szczecin, 19 Piastow Ave., 70-310 Szczecin, Poland; rafal.grzejda@zut.edu.pl; 3Laboratory of Biocomposite Technology, Institute of Tropical Forest and Forest Product (INTROP), University Putra Malaysia, Serdang 43400, Malaysia; 4Aerospace Malaysia Innovation Centre [944751-A], Prime Minister’s Department, MIGHT Partnership Hub, Jalan Impact, Cyberjaya 63600, Malaysia; 5Institute of Mechanical Engineering, Faculty of Mechanical Engineering, Bialystok University of Technology, 15-351 Bialystok, Poland; w.tarasiuk@pb.edu.pl; 6Faculty of Technical Sciences, University of Warmia and Mazury in Olsztyn, 10-719 Olsztyn, Poland; rychter@uwm.edu.pl

**Keywords:** phased array ultrasonic testing (PAUT), non-destructive testing (NDT), ultrasonic thickness gauge (UTG), multi-bolted connections, flat bottom hole, elevated temperature

## Abstract

Petrochemical plants use on-stream inspection often to detect and monitor the corrosion on the equipment and piping system. Compared to ultrasonic thickness gauging and pulse-echo A-scan, phased array corrosion mapping has better coverability and can scan a large area to detect general and localized corrosion. This paper’s objective is to obtain documentary evidence for the accuracy of corrosion detection from 30 °C to 250 °C on A36 low-carbon steel by carrying out simulation experiments every 10 °C step. A minimum of three sets of phased array corrosion mapping data in each temperature were collected to study and evaluate the detectability. The data evidence could enhance the confidence level of the plant’s end users in using phased array mapping in the future during inspections. The experiments were found to be insufficiently thorough despite addressing the initial concerns, leaving more area for discussion in further studies, such as expanding the investigation to thicker carbon steel, stainless steel, and wedge materials.

## 1. Introduction

Existing petrochemical plants use on-stream inspection often to detect and monitor corrosion on the equipment and piping systems, which are typically built with multi-bolted connections [1,2]. Corrosion on the pipeline’s walls can affect the connections’ deformation and change the bolts’ tension, similar to how pipes bend [3]. Any such change in bolt tension can in turn cause dangerous leakage of the fluid being transported, which is why adequate bolt tension is important in flanged pipe connections [4,5].

On-stream inspection is a plant’s maintenance system, which monitors and assesses the plant’s operation; the benefit of on-stream inspection includes improving the turnaround time because some essential problems may be monitored while the plant operates.

The current on-stream inspection technique to detect piping corrosion is ultrasonic thickness gauging (UTG), but it may not detect localized corrosion [6].

Phased array ultrasonic testing (PAUT) can scan a large area to detect general and localized corrosion [7], but it still faces the same ultrasonic inspection limitation that exceeds the standard recommended ultrasonic testing temperature of 52° [8] when carrying out on-stream corrosion monitoring at some piping operating temperatures greater than 200 °C.

So far, there is still uncertainty in applying phased array corrosion mapping on high-temperature material surfaces. It can be seen that some petrochemical plant end users are concerned about the temperature that the detection can reach. Some practitioners will answer affirmatively, saying that a specific temperature can be achieved. Still, a further question is how the sensitivity will change at such a high temperature. Currently, there is no detailed answer, mainly because there are no apparent reliable data.

To the best of the authors’ knowledge, currently, only Turcu et al. conducted phased array corrosion mapping up to 150 °C using a dual linear probe, and they reported that the challenges faced included the ultrasound velocity change when the temperature rises and the type of couplant and scanner selection, but with limited results, discussion, and presentation [9].

This paper aims to obtain documentary evidence for phased array corrosion mapping detectability from 30 °C to 250 °C on low carbon steel by conducting simulation experiments every 10 °C step. The data evidence could enhance the confidence level of petrochemical plants’ end users in using phased array corrosion mapping in the future during on-stream inspections to reduce petrochemical plants’ downtime due to inspections.

Detailed experimental sequences are discussed in the methodology section, including identification of the test specimen materials, test specimen design and fabrication, elevated temperature simulation, test equipment selection, and testing procedure.

The samples used in the test were A36 low-carbon steel plates with a length of 200 mm, width of 100 mm, thickness of 15 mm. Machining slots of different depths represented general corrosion, and flat-bottom holes of different depths and diameters represented localized corrosion.

At least three sets of phased array corrosion mapping data were collected from the experimental tests at 30 °C to 250 °C, with a difference of 10 °C in each step. The carbon steel plate was heated by a heating element and monitored with a thermocouple and hand-held thermometer with temperature accuracy of ±0.5 °C.

## 2. Literature Review

Considering that the readers of this article may come from different fields, the articles discussed in the literature review section introduce the background of this experiment in detail, including the current plant maintenance strategies and the damage mechanism, especially focusing on corrosion and localized corrosion, including an introduction to the current inspection techniques’ limitations.

One of the most crucial approaches is condition-based maintenance (CBM). CBM policy is based on data collected through condition monitoring. CBM primarily aims to prevent breakdowns and malfunctions by monitoring essential equipment component parts [10].

The purpose of condition monitoring is to inspect various types of petrochemical equipment, such as storage tanks, pressure vessels, and piping systems [11], without shutting down the plant. Many different types of NDT methods/techniques can perform condition monitoring, but four factors should be considered when choosing an inspection method: 1. the kind of damage to the targeted mechanism; 2. the size of the targeted defect; 3. the location of the defect; and 4. the inspection method’s sensitivity and limitations [12]. For example, if the targeted defect is located in the inner shell of the equipment with no external access or operating temperature, the inspection results may be invalid. The condition monitoring location (CML) is the place chosen to assess the remaining thickness on the potential corrosion location regularly, often by ultrasonic thickness measurements to detect the wall loss.

Wagh demonstrated the advantage of employing RBI evaluations, rather than the standard remaining life calculation, to evaluate the remaining life of above-ground storage tanks and process piping in a petrochemical plant [13]. The RBI was calculated based on the type of incident and the consequences if equipment fails, as well as the likelihood that the incident will occur; for example, the likelihood of new equipment failure until some period may be negligible, so no shutdown inspection for the equipment is required during this period.

Inspection manager computer software based on the RBI methodology has been created to assist in selecting response actions to unplanned events resulting in the release of hazardous materials. The RBI approach is frequently used to identify important equipment for which inspection will provide the greatest benefit in terms of reducing operational risk. The costs connected with accidents are complex, especially when human life and deformed environments are involved, presenting a substantial obstacle in the decision-making process [14].

The damage mechanism is chosen based on several parameters, including the material composition of the equipment, the nature of the fluid treated or stored in the equipment, the surrounding processing environment, and other elements that influence the screening of damage criteria. By considering these elements, one can determine the exact types of damage processes that are likely to occur in the equipment. This information is required for successful inspection and maintenance procedures to prevent equipment deterioration or failure [15]

Liquid hydrocarbons, dissolved gases, water, and salts compose crude oil. This oil exists in the form of an emulsion, with water droplets scattered inside the continuous hydrocarbon phase. On the other hand, natural gas is a mixture of hydrocarbons, nitrogen, carbon dioxide, sulfur dioxide, water, and trace amounts of mercury, organic acids, and inert gases. Corrosive substances, such as CO_2_, H_2_S, H_2_O, mercury, and organic acids, can cause metal corrosion in the oil and gas sector throughout various stages of natural gas production, separation, processing, transportation, handling, and storage [16].

Carbon dioxide generates a type of corrosion known as sweet corrosion. Sour corrosion happens when H_2_S causes corrosion. O_2_ corrosion is referred to as oxygen corrosion [16]. H_2_S can cause general and pitting corrosion, as well as hydrogen attack. Cracks and blisters are indicators of further growth. Human death and environmental damage can arise from the leakage of H_2_S into the environment through fissures.

The appearance of oxygen corrosion is pitting. Oxygen corrodes carbon steel, low alloy steels, copper, and alloys. However, oxygen is required to keep protective oxide coatings on stainless steel, titanium, and aluminum in place.

The corrosion rate increases as the temperature rises. On the other hand, the solubility of hazardous gases diminishes with increasing temperatures. The temperature and corrosion rate have a complicated relationship. When the temperature reaches a particular point, the corrosion rate accelerates. The corrosion rate decreases when the solubility of corrosive gases in aqueous solutions diminishes beyond a certain temperature. When the temperature is low, H_2_S speeds up corrosion. These phenomena are particularly dangerous in view of the turbulent nature of the flow occurring under certain conditions in pipelines [17,18].

According to Wang and Yang, corrosion accounts for almost 80% of overall petrochemical plant failures. They emphasized that, for every 10 °C increase in temperature, the corrosion rate increases by 1–3 times. Furthermore, increasing pressure can increase the solubility of corrosive gases, hastening the corrosion process. These studies illustrated the importance of temperature and pressure in petrochemical plant corrosion rates [19]. Managing and reducing corrosion through appropriate material selection, protective coatings, and corrosion control techniques are critical to ensuring the integrity and reliability of petrochemical equipment and infrastructure.

Corrosion is a localized electrochemical oxidation and reduction reaction that occurs on metal surfaces. It consists of an anode, a cathode, and an aqueous solution or electrolyte containing positively and negatively charged ions with conductivity. Electrons are moved from the metal surface to another site during the electrochemical corrosion process, resulting in slow deterioration and eventual failure of the metal [20].

Internal corrosion is caused by gases or liquids that are stored or moved within the system. Corrosion can occur under both anaerobic (oxygen-free) and aerobic (oxygen-containing) conditions when metals are continuously exposed to these fluids [20]. This continuous exposure to fluids causes corrosion, jeopardizing the integrity and function of metal structures. Preventive measures, such as corrosion-resistant materials, protective coatings, and good maintenance practices, are required to limit the effects of corrosion and preserve the longevity of metal components in various industrial contexts.

Carbon steel corrosion is a serious problem in many chemical processing industries, including fire protection networks. Despite the advent of alternative piping materials, carbon steel remains a popular choice for transporting chemicals, acids, hydrocarbon products, and water due to its adaptability and low cost [21].

Corrosion can cause a gradual decrease in the wall thickness of carbon steel pipes over time. This reduction in thickness causes greater hoop stress, both throughout the pipe and in specific regions. As the hoop stresses interact mechanically with the pipe wall, they can cause expansion, buckling, deformation, and even rupture.

While welding defects, stress corrosion cracking, and hydrogen cracking can all cause significant damage to pipes, they have not been shown to affect pipe wall thickness directly. These concerns, however, can contribute to total pipe degradation and jeopardize the integrity of carbon steel piping systems. To limit the impact of corrosion and guarantee the safe and dependable operation of carbon steel piping networks, preventive measures, such as regular inspection, maintenance, and corrosion management techniques, must be used [22].

General corrosion, also known as uniform corrosion, happens when a metal loses electrons due to electron flow to the cathode of the same metal. It is a common type of corrosion that uniformly affects the entire surface of the metal. Corrosion in general can occur in a variety of settings, including those containing chlorides, sulfides, and carbon dioxide, and it can result in faster wear of various parts of machinery and equipment [23,24].

The presence of chlorides, sulfides, and carbon dioxide in the crude distillation overhead system can lead to the development of general corrosion. These corrosive substances affect the system’s metallic structure, slowly weakening the metal. The general corrosion assault might cause a loss in the thickness of the metallic structure over time [25]. Preventive methods, including using corrosion-resistant materials, applying protective coatings, frequent inspections and maintenance, and appropriate corrosion control strategies, can be applied to mitigate general corrosion in such systems. The impact of general corrosion can be reduced by following these steps, ensuring the safe and efficient operation of the crude distillation overhead system.

Pitting corrosion, also known as localized corrosion or pinhole corrosion, is a type of corrosion with a small diameter but excessive depth. Pits arise of a microscopic size and cause metals and alloys to fail via perforation and stress corrosion cracking, among other failure modes. Pitting can be classified into three stages: nucleation, growth and passivation, and sound stage pitting. Pitting is influenced by the material microstructure, chemical content, grain size, temperature, corrosive media, and pH [26]. Localized corrosion is more damaging than uniform corrosion, accelerating the initiation of corrosion leaks [27]. Corrosion can lead to corrosion cracking or stress corrosion cracking, which can break the piping material [28,29].

Z.G. Zhang et al. presented a nondestructive testing (NDT) approach for performing inspections during plant operations in a high-temperature environment. Thermal imaging, ultrasonic testing, and pulsed eddy currents are examples of high-temperature capable techniques [30]. The use of UTG to identify plant corrosion during plant operation can check environments of up to 200 °C. However, different technicians’ and equipment’s uncertainty may not be accurate at the same UTG point.

Ultrasonic testing is most commonly applied to on-stream inspection compared to other non-destructive tests mainly because of the lightweight equipment that provides greater accessibility in the complex petrochemical plant’s environment. The test only requires accessing one side of the material surface, and it is non-radiation hazardous and can be performed with other trade workers around simultaneously [31]. A non-contact thickness measurement was introduced by S. Li et al. that could perform point measurements in up to 400 °C [32].

Compared to other ultrasonic inspection techniques, such as ultrasonic thickness gauging and pulse-echo A-scan, phased array corrosion mapping has better coverability [6].

Turcotte, Rioux, and Lavoie investigated tank corrosion mapping examinations using conventional UT, phased array ultrasonic testing (PAUT), and 3D scanners. Unlike conventional UT beams, which only report one thickness at a time, phased array scans can produce a variety of thicknesses. One of the primary advantages of employing a phased array for corrosion mapping is that it functions in the same way as a conventional UT probe array, with all probes aligned with precise overlap and working concurrently. The size and number of probe elements, in addition to frequency, are the primary drivers.

In terms of time and coverage, PAUT has an advantage over conventional ultrasonic thickness measurement. With conventional ultrasonic thickness measurement, it takes 2.7 h to cover a 100 mm × 100 mm grid with a 1 mm grid, assuming 1 second for each measurement step. The same area can be covered in a matter of seconds with phased array corrosion mapping [33].

## 3. Methodology

The quantitative experiment began with handheld laser-induced breakdown spectroscopy to verify the low-carbon steel material grade. Subsequently, designing the test specimen with detailed consideration and preparing the test specimen by machining occurred. The simulation method to raise the material temperature and the testing procedure, including the equipment and accessories selection, are explained in this section.

### 3.1. Test Specimen

A test specimen constructed of low-carbon steel was employed in the study. Because of its advantageous qualities, this material grade is widely used in the petrochemical industry. The test specimen measured 200 mm in length, 100 mm in width, and 15 mm in thickness. However, the original mill certificate or documentation that would normally identify the precise low-carbon steel grade of the item was lost during the experiment.

The types of technical operations were taken from the production engineer’s dictionary. A handheld laser-induced breakdown spectroscopy (LIBS) approach was used to evaluate the material grade and certify that it corresponded to A36 low-carbon steel, as shown in Table 1.

This study used the SolidWorks CAD system (2021 version 29) to produce the test specimen for the experiment [34]. The authors created a three-dimensional (3D) model of the test specimen using computer-aided design software [35]. A 3D drawing prepared in a 1:1 ratio to ensure correctness and effective communication with the CNC milling process [36].

Therefore, both general corrosion and localized corrosion were considered in the specimen design to simulate the wall loss. Instead of a side drill hole [37], general corrosion was represented by a machining notch with 5 mm in width and varying depths of 2 mm, 4 mm, 6 mm, and 8 mm. In comparison, localized corrosion is represented by flat bottoms holes (FBHs) [38] and varying diameters of 20 mm, 15 mm, 10 mm, and 5 mm from rolls with different depths of 3 mm (20% wall loss) and 6 mm (40% wall loss) for calibration and performing of experimental tests [39]. Figure 1 provides the test specimen design with detailed dimensions, and Figure 2 contrasts the specimen 3D view and the fabricated test specimen by machining.

### 3.2. Elevated Temperature Simulation Method

To successfully apply phased array corrosion mapping for on-stream inspection, the experiment was conducted at temperatures ranging from 30 °C to 250 °C to simulate plant operating conditions. The carbon steel plate was placed on a ceramic pad heater element to heat from 30 °C to 250 °C, with a difference of 10 °C in each step, and it was monitored with a thermocouple and hand-held thermometer with temperature accuracy of ±0.5 °C.

An experiment across a temperature range of 30 °C to 250 °C was carried out to assess the possibility of using phased array corrosion mapping for on-stream inspection in plant operating circumstances. The goal was to replicate the real operational conditions of the plant where the inspection would take place.

To accomplish this goal, a carbon steel plate was placed on top of a ceramic pad heater element. The heating element aided in the progressive rise in temperature from 30 °C to a high of 250 °C. The temperature was raised in 10 °C increments, allowing for a thorough evaluation of the phased array corrosion mapping technique at various temperature points.

Thermocouples were strategically placed on the carbon steel plate’s top, middle, and bottom sections to ensure exact temperature monitoring. Throughout the heating process, these thermocouples delivered precise temperature readings. A hand-held thermometer with a high degree of precision of 0.5 °C was also employed as an additional means of temperature verification.

The authors examined the effectiveness and dependability of the phased array corrosion mapping technique in a range of operational temperatures by running the experiment under controlled temperature circumstances and closely monitoring the temperature changes. These data are critical for verifying the technique’s applicability and performance in real-world settings with changing temperatures encountered during plant operations.

### 3.3. Phased Array Corrosion Mapping

PAUT is an ultrasonic inspection technique that can simultaneously send multiple elements, just like scanning with multiple conventional ultrasonic probes [40]. The advantage is not only the coverage of 30 mm in width with one scan but also the possibility of providing three dimensions of defect information [41].

There are several types of software used depending on the phased array equipment manufacturer, but most manufacturers commonly use displays consisting of A-scans (conventional ultrasonic testing display with echo), B-scans (to show a side view and display the inspected length), C-scans (to show a top view and display the inspected length), and S-scans (to show a cross-section view or so-called sectorial scan) [42].

The multiple views of the collected data enable phased array interpreters to identify and justify the defect type and location easily.

A difference between this experiment and the general phased array corrosion mapping is that the temperature of the surface temperature of the detected object was higher than the standard recommended temperature. Some challenges were encountered because the experiment was not based on commonly known operating standards.

The first obstacle was the possibility of damaging the probe’s piezo-element at extremely high surface temperatures. The probe’s piezo-element may be subjected to thermal stress as temperatures rise, resulting in degradation or even complete failure [43,44]. It was necessary to consider this point and devise methods to reduce the influence of high temperatures on the probe’s performance.

The nature of the ultrasonic couplant commonly utilized in phased array corrosion mapping posed the second challenge. The traditional couplant is water-soluble and evaporates quickly as the temperature rises. When the surface temperature of the test object exceeds 100 °C, the functionality of the couplant is disrupted. Loss of the couplant can result in insufficient coupling between the probe and the test object, resulting in lower signal transmission and poor inspection accuracy [45].

Addressing these issues necessitated the development of novel solutions that were suited to the individual experimental conditions. Alternative materials or protective coatings, for example, might be investigated to improve the thermal resistance of the probe’s piezo-element, preserving its integrity even at high temperatures. Furthermore, selecting a suitable couplant or developing specialized high-temperature couplants capable of withstanding the extreme temperatures experienced during the inspection process would be required to maintain dependable coupling between the probe and the test object.

This study aimed to build a framework for conducting phased array corrosion mapping at higher surface temperatures, allowing for more effective on-stream inspection in industrial environments with heightened temperatures by recognizing and overcoming these limitations.

Additionally, longitudinal velocity is obtained by measuring the round-trip echo from the known material thickness [46]. An earlier experiment that successfully addressed the issue of how the temperature of the object being detected might affect the ultrasonic velocity was employed in this study. The earlier experiment verified the theoretical velocity calculation, including the heat-conduction coefficient K [47]. Figure 3 shows the longitudinal velocity values between 30 °C and 250 °C. After considering the aforementioned issues and conducting several trials, a preferable testing combination was ultimately discovered.

The testing uses a high-temperature-resistant wedge [48] and focuses on limiting contact time during detection, while the phased array probe uses a general-purpose 5L64 probe [49]. The water-soluble couplant is also replaced with high temperature-resistant oil [50].

A schematic diagram of the phased array corrosion mapping experimental setup is shown in Figure 4 below, where the heat treatment equipment is used to heat the specimen to a specific temperature required for the experiment, and the PAUT equipment collects the data and further transmits them to a computer for interpretation.

The process flow of this experiment can be summarized in several steps:Determine the test specimen’s base material grade and validate it with portable laser-induced breakdown spectroscopy. This stage confirms that the material composition, such as the A36 low-carbon steel grade, satisfies the needed criteria;Select the temperature range to examine and refer to the velocity values. It is well understood that velocity decreases with increasing temperature, and a reference list of velocity values ranging from 30 °C to 250 °C is provided for comparison during the experiment [51];Heat the test specimen to the specific experimental temperature. At the same time, use the Figure 3 velocity value for the specific temperature to the calibrate PAUT equipment;The temperature of the test specimen is monitored using thermocouples strategically positioned. Temperature values are checked and confirmed with a thermal meter to ensure accuracy and reliability;To obtain data from the machined slots and FBH, perform the PAUT test on the heated test specimen. The phased array probes are precisely positioned and aligned to achieve excellent coverage and data capture;Transfer the test data collected to a computer for interpretation and analysis. Specialized software is employed to process the data and visually represent the corrosion mapping results;Once the specific temperature data are collected, the temperature can be increased or decreased to another temperature by a heat treatment machine. For example, for the three sets of 110 °C data collected, the next step is to heat the specimen to 120 °C;Perform the sensitivity check by setting the second back wall echo at 80% full-screen height (FSH) in A-scan mode and increase or decrease the decibel (dB) level as the temperature changes [52];Repeat steps 5–7 until at least three data sets are gathered in each temperature range. This repeating enables data validation and statistical analysis to guarantee that the results are consistent and reliable.

## 4. Results and Discussion

The corrosion mapping data collected from the PAUT device are transferred to a computer, where they are interpreted by software. The advantage of this testing technique is that different depths can be referenced by color codes for direct visual comparison. In addition, additional information, such as the indicated extraction size measurable by plotting or the extraction depth of a specific area, can be obtained by simply moving the cursor to a specific location. However, visual information allows for quick identification of different depths.

The results from the experiment can be classified into five separate stages based on the temperature range. The first step covers the temperature range of 30 °C to 60 °C. The velocity of the ultrasonic waves decreased as the temperature increased throughout this period. However, the decibel (dB) levels remained constant, resulting in duplicated data with slight variance. One of the experiment’s most significant features was successfully identifying the machined slots that reflected general corrosion. These slots were created at various depths, including 2 mm, 4 mm, 6 mm, and 8 mm. They had distinct edges that were seen in the corrosion mapping data. To aid in the visualization and interpretation of the extent of corrosion, each depth was assigned a different color. Some colors reflected like a poor-resolution image because of reflections of the actual signal, whereas the bottom scale could be clearly identified.

These findings illustrated the utility of PAUT corrosion mapping in identifying and characterizing corrosion at various depths. The unique colors provided for each depth level improved the visual depiction of corrosion features, allowing for exact assessment and analysis of the extent of corrosion.

The two rolls of different-depth FBHs that represented localized corrosion could be identified clearly in the 6-mm depth (40% wall loss) roll in all the diameter sizes, the 3 mm depth (20% wall loss) FBHs noted for 10-mm, 15-mm, and 20-mm diameters could be identified clearly in sizes, but in the 5-mm diameter FBHs, the data could identify the indication but were unable to present the round shape clearly. Figure 5 shows the phased array corrosion mapping data image at 50 °C.

The experiment’s second stage covered a temperature range of 60 °C to 100 °C. During this stage, increasing the decibel (dB) level as the temperature rose was required to achieve an acceptable sensitivity level for corrosion detection. The data obtained in this stage were identical to the preceding group in that machined slots and flat-bottom holes (FBHs) were discovered. However, only the 3-mm depth (representing 20% wall loss) FBHs did not clearly display the shape of the 5 mm diameter FBHs. Figure 6 shows the phased array corrosion mapping data image at 100 °C.

In the third stage, due to the elevated temperature settings, particular measures had to be taken, which covered the temperature range of 100 °C to 140 °C. The scanning technique had to be carried out with greater caution because the couplant used in the method began to vaporize, causing smoke to appear. This smoke created an impediment and challenge in gathering correct data. To resolve these issues, it was critical to reduce the scanning time and remove excessive couplant to keep the surface of the material dry after scanning, thus eliminating the interference of smoke with subsequent work. Shortening the scanning procedure might reduce heat transfer from the wedge to the probe’s piezo-element, lowering the risk of probe damage. Figure 7 shows the data from 110 °C to 140 °C, which are not too far from the 100 °C data.

Similar to the second stage, decibel level changes were required during the third stage to ensure ideal sensitivity levels. The observed velocity changes necessitated fine-tuning of the decibel settings. Despite the difficulties created by the elevated temperatures, the machined hole representing general corrosion could be clearly detected at all possible specified depths. The existence of a distinct edge in the data allowed for reliable corrosion detection and characterization. It was noted, however, that the 3-mm depth (representing 20% wall loss) FBH data still had difficulty in showing the whole round shape of the 5-mm diameter FBHs.

In the subsequent stage, with temperature ranging from 150 °C to 180 °C, as Figure 8 shows, the machined slot images boundary lines at different depths begin to blur, and the display color of the 2-mm slot was the same as the color of the material surface, making it difficult to distinguish. In contrast, the 5-mm diameter FBHs could not be shown as round before showing signs of improvement.

The last stage of data was from 190 °C and to 250 °C, with the data collection conditions more severe in this temperature range. The heat could be felt during scanning even while wearing insulated gloves. This procedure required more accurate data acquisition to reduce the probe contact time. This stage also had to have higher decibel values to maintain good imaging. The machined slot images in this stage could be identified except for the 8-mm slot depth images unable to show the 5-mm slot width. The two rolls of different-depth flat bottom holes that represent localized corrosion could be identified clearly in both the 6-mm depth (40% wall loss) and 3-mm depth (20% wall loss) rolls, except for the middle of the diameter of 6-mm roll FBHs, which showed some uncaptured image shapes. Still, it did not affect the overall result. Figure 9 shows the phased array corrosion mapping data image at 220 °C.

Table 2 further compares the detected depths from 30 °C to 250 °C with the design depth. At 240 °C, C4 shows the largest deviation, 0.78 mm or less than 1 mm, and most of the other data show deviations of less than 0.5 mm. Summarizing the overall deviations at each temperature, Figure 10 shows that the largest deviations were found at 90 °C and 230 °C. Conversely, the smallest relative deviation was from 100 °C to 140 °C.

## 5. Conclusions

This study aimed to collect a large amount of data to explore the efficacy of phased array corrosion mapping on low-carbon steel across a wide range of temperatures. A total of 69 data sets were methodically gathered, ranging from 30 °C to 250 °C at 10 °C intervals. These comprehensive data sets were critical to meeting the project’s objectives and providing valuable insights for this study.

The results of the phased array corrosion mapping experiments on low carbon steel at temperatures ranging from 30 °C to 250 °C revealed the detectability of corrosion in the given temperature range.

The novelty of this study is that it documents the capability and detectability of corrosion mapping during petrochemical plant operations at temperatures less than 250 °C.

Although UTG is not the best technique for detecting localized corrosion, phased array corrosion mapping has not been widely used in the industry during on-stream inspection due to the lack of clear codes, standards, specifications, and guideline support. The results of this paper could be brought to the end user and maintenance vendor for alternative options. There is no need to wait for turnaround maintenance to inspect the suspected corrosion pipeline to reduce overall costs and increase plant integrity.

This article could be used as communications material between academia and industry to assist industry in discussing and resolving real-world issues. It will continue to expand research in the direction of future work to achieve contributions to the industrial field.

## 6. Future Work

The study’s inadequacies or gaps present additional research and investigation opportunities. By filling these gaps, researchers can better understand the effectiveness and limitations of phased array corrosion mapping with various temperature situations, material compositions, and equipment configurations. This article lays the groundwork for future research and urges the scientific community to participate in discussions and collaborative efforts to develop the field of ultrasonic inspection and corrosion mapping techniques, particularly in the field of multi-bolted jointed pipelines.

The experiment in this study worked with a base metal thickness of 15 mm. Although this thickness was adequate to reflect the thicknesses of common pipe walls, it is important to note that thicker base metals may be used in subsequent tests to broaden the application area of phased array corrosion mapping’s use in petrochemical plants. As a result, the technique might be applied to a broader range of equipment in such facilities [53].

Future expansion of the size and depth range design of slots and FBHs, as well as detailed studies of accuracy or variation in temperature, e.g., plotting of standards/deviations of individual defects, are good directions toward more comprehensive studies.

The use of thicker base metals in future investigations is significant because of their effects on ultrasound attenuation. The attenuation of ultrasonic waves becomes more pronounced as the thickness of the material increases. As a result, exploring the combination of greater material thickness and high temperatures would be an ideal future research direction. It could help us to more thoroughly understand the ultrasonic attenuation phenomenon by investigating thicker base metals and their behaviors under high-temperature settings. This knowledge would help to build more precise and dependable inspection strategies, allowing for effective corrosion mapping in various petrochemical plant equipment.

Extending the experiment to encompass thicker base metals and higher temperatures would increase the utility and versatility of phased array corrosion mapping in industrial settings. It would provide useful insights into the behaviors of different thicknesses of materials and aid in optimizing the inspection procedures for enhanced corrosion detection and monitoring.

Although carbon steel is the most widely used material in the petrochemical industry, other base materials must be addressed. For example, stainless steel can be more resistant to corrosion and high pressure and relatively error-free. Otherwise, the consequences will be rather severe. Stainless steel is also the base material most likely to encounter stress corrosion cracking, which would be beneficial for use in monitoring stream corrosion [54] and sliding contact temperature [55].

The material of the wedge is also a good direction for discussion. If it can be used without any risk of high temperatures and be in contact for a longer time, it will result in greater convenience and reduce the overall inspection time.

The observation that 5-mm FBHs cannot show a complete image at low temperatures but become more clearly reflected at high temperatures implies that the objects being tested have thermal tension. This discovery raises the possibility of additional discussion and investigation.

A major problem is the inability to detect minor faults when pipes are at room temperature during shutdown inspections. This inability means that specific defects or imperfections may be more difficult to detect and diagnose under normal working settings. This fact emphasizes the importance of inspecting at higher temperatures, mirroring actual working conditions and providing a more accurate depiction of the material’s behavior and the existence of flaws. Further investigations and discussion of the impact of thermal tension on ultrasonic testing and the detection of tiny defects could yield useful insights and advances in corrosion mapping techniques.

In this vein, it is critical to discuss the current research findings with petrochemical plant end users to solicit their feedback and suggestions for the prospective use of the inspection design in real-life situations. Obtaining their comments would be beneficial for determining the feasibility and practicability of incorporating research findings into their ongoing projects. Engaging with end users will also provide insights into the unique difficulties that may develop in the site context, allowing for further study extension.

Researchers can better understand the industry’s objectives and restrictions by partnering with petrochemical plant end users, leading to the development of specialized inspection methodologies that are more effective and consistent with real-life circumstances. This collaboration could promote knowledge sharing and the development of best practices, ultimately leading to breakthroughs in inspection methodologies and equipment selection.

## Figures and Tables

**Figure 1 materials-16-05297-f001:**
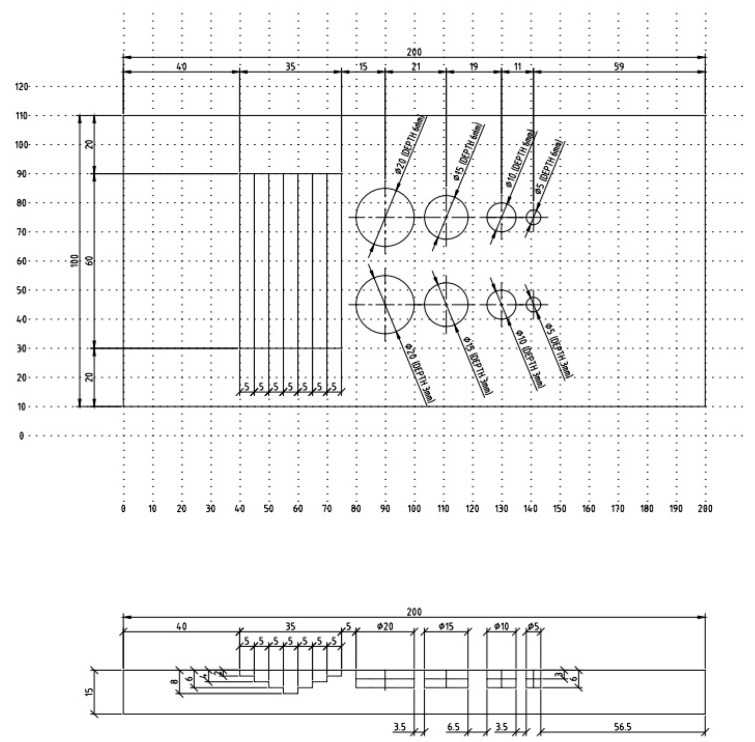
Schematic of the test specimen design with detailed dimensions.

**Figure 2 materials-16-05297-f002:**
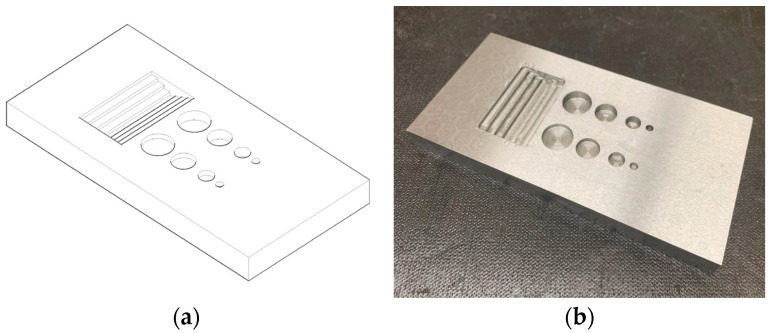
(**a**) Test specimen 3D view; (**b**) fabricated test specimen.

**Figure 3 materials-16-05297-f003:**
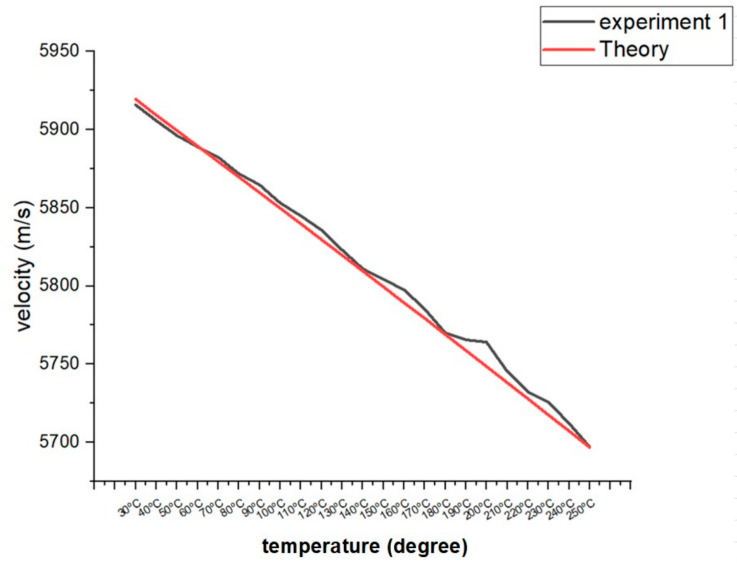
Longitudinal velocity values between 30 °C and 250 °C in carbon steel.

**Figure 4 materials-16-05297-f004:**
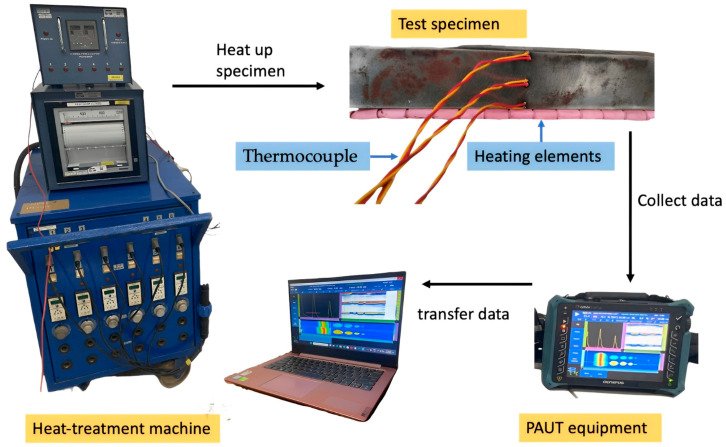
Schematic of the experimental setup for performing phased array corrosion mapping.

**Figure 5 materials-16-05297-f005:**
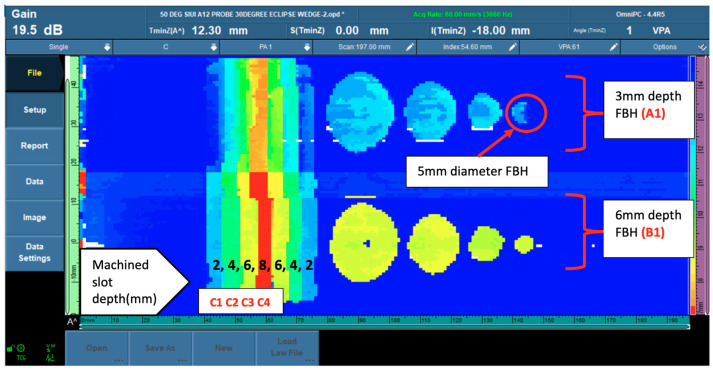
PAUT corrosion mapping data at 50 °C.

**Figure 6 materials-16-05297-f006:**
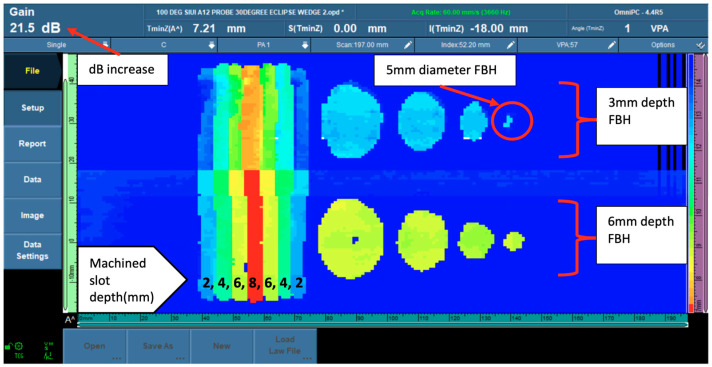
PAUT corrosion mapping data at 100 °C.

**Figure 7 materials-16-05297-f007:**
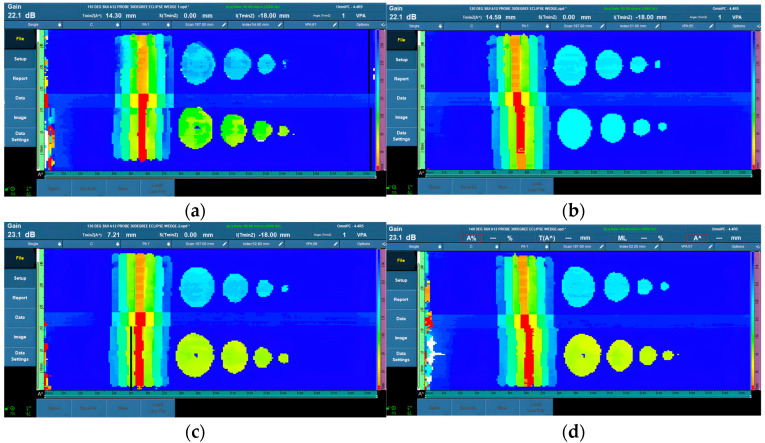
PAUT corrosion mapping data from 110 °C to 140 °C. (**a**) data at 110 °C; (**b**) data at 120 °C; (**c**) data at 130 °C; (**d**) data at 140 °C.

**Figure 8 materials-16-05297-f008:**
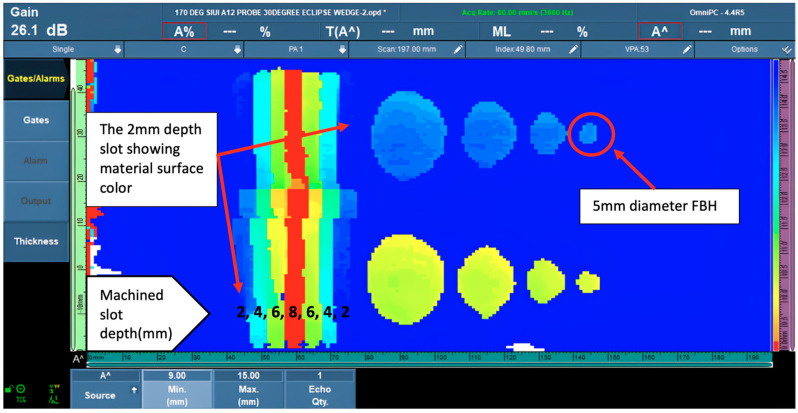
PAUT corrosion mapping data at 170 °C.

**Figure 9 materials-16-05297-f009:**
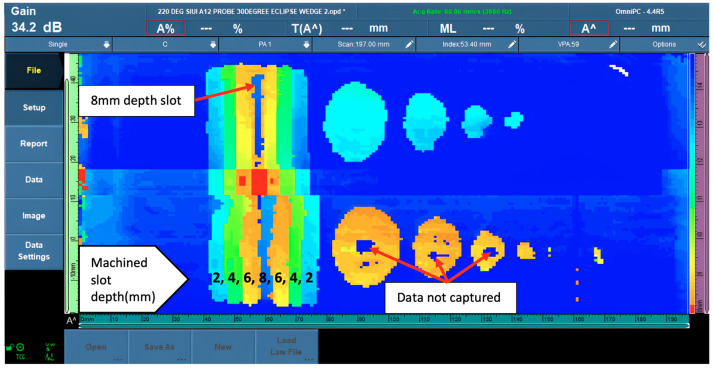
PAUT corrosion mapping data at 220 °C.

**Figure 10 materials-16-05297-f010:**
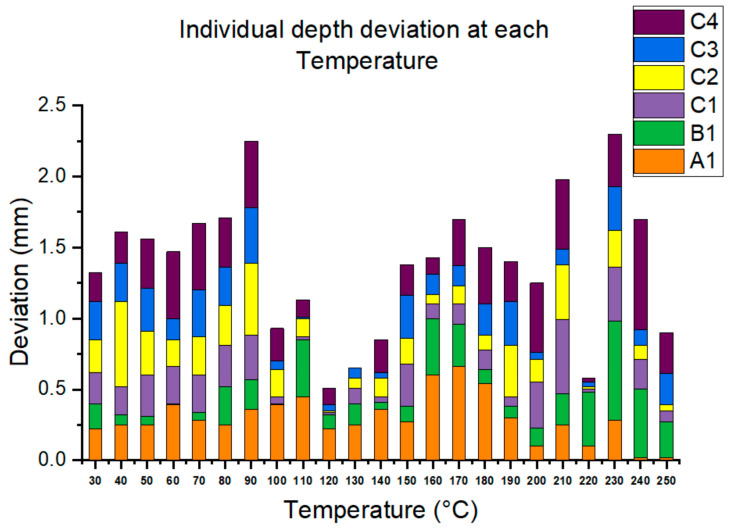
Overall depth deviation detected at each temperature.

**Table 1 materials-16-05297-t001:** Material chemical composition from LIBS.

Element	Carbon (C)	Iron (Fe)	Silicon (Si)	Phosphorus (P)	Sulfur (S)
Content	0.146%	97.79%	0.275%	0.018%	0.021%

**Table 2 materials-16-05297-t002:** Comparison of design depths and actual detected depths.

Design Depth	A1 (3 mm)	B1 (6 mm)	C1 (2 mm)	C2 (4 mm)	C3 (6 mm)	C4 (8 mm)
Temperature	Detected Depths					
30 °C	2.78	6.18	2.22	4.23	6.27	8.20
40 °C	2.75	6.07	2.20	4.60	6.27	8.22
50 °C	2.75	6.06	2.29	4.31	6.30	8.35
60 °C	2.61	6.01	2.26	4.19	6.15	8.47
70 °C	2.72	6.06	2.26	4.27	6.33	8.47
80 °C	2.75	6.27	2.29	4.28	6.27	8.35
90 °C	2.64	6.21	2.31	4.51	6.39	8.47
100 °C	2.61	6.01	2.05	4.19	6.06	8.23
110 °C	2.55	5.60	2.02	4.13	6.01	8.12
120 °C	2.78	6.10	2.02	4.01	6.04	8.12
130 °C	2.75	6.15	2.11	4.07	6.07	8.00
140 °C	2.64	5.95	2.04	4.13	6.04	8.23
150 °C	2.73	6.11	2.30	4.18	6.30	8.22
160 °C	2.40	5.60	1.90	3.93	5.86	7.88
170 °C	2.34	5.70	1.86	3.87	5.86	7.67
180 °C	2.46	5.90	1.86	3.90	5.78	7.60
190 °C	3.30	6.08	2.07	4.36	6.31	8.28
200 °C	3.10	6.13	2.32	4.16	6.05	8.49
210 °C	2.75	6.22	2.52	4.39	6.11	8.49
220 °C	2.90	5.62	2.02	4.02	5.97	8.03
230 °C	2.72	6.70	2.38	4.26	6.31	8.37
240 °C	2.98	6.48	2.21	4.10	6.11	8.78
250 °C	2.98	6.25	2.08	4.04	6.22	8.29

## Data Availability

All data generated or analyzed during this study are included in this article.

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
