# Peer review of "Experimental Investigation on the Corrosion Detectability of A36 Low Carbon Steel by the Method of Phased Array Corrosion Mapping"

_materials, 2023, doi:10.3390/ma16155297_

Round 1

Reviewer 1 Report

In this paper, the detectability of phased array under different temperature conditions is studied for the corrosion defects of A36 mild steel. The establishment of a large number of temperature detection data sets provides a certain reference for the subsequent evaluation of the reliability of high temperature detection.

1 There are many factors affecting the experimental results. As mentioned in the paper, when the temperature reaches 100-140 °C, the coupler evaporates and produces smoke. If this has a great impact on data collection, can a coupler with higher temperature resistance and no smoke be used instead of the experiment?

2.It is mentioned in the paper that the decibel level needs to be changed in the second and third stages to ensure the detection sensitivity level.  Please describe in detail.

3. This paper only conducts corrosion defect detection on A36 low carbon steel, and the detection effect is obviously different under different material composition and equipment configuration. May I ask how to improve it in the next step, so that the research in this paper has certain generalization ability?

Reviewer 2 Report

The authors are presenting a work to investigate the corrosion mapping by using ultrasonic phased array. Documentary evidence on the accuracy of corrosion detection at different temperature is obtained to verify the accuracy of the corrosion detection system. This paper is not an academic article, seems verification experiment. The work is lacked some key experiment, content and discussion. The novelty is very confused, the content is hard to understand. Is the experimental set-up self-designed or from the NDT company? Is the phased array proposed by the authors, or just bring from the instrument? The new contribution and novelty were not provided and explained clearly. Not much new knowledge about corrosion detection method can be gained.

Besides these basic arguments, there are a number of further points that would need to be considered:

1)     The content of the paper is not understood. The author takes much space to introduce the introduction. Is it a review paper?

2)     Similarly, four pages is occupied to introduce the Methodology. The authors introduce CAD3D drawing and CN machining fabrication of the sample preparation in detail. It is no any importance. The authors just need to briefly explain the sample preparation, experimental process and the experimental set-up.

3)     The author did not provide Figure and table to illustrate the results. Some capture results from the instrument is given in the text. The authors need to reprocess the data in Figs 3-6.  

4)     The principle of phase array mapping should be explained clearly in the section of methodology.

The work is just an experimental report, not a strict academic work. Based on this overall impression, I recommend to reject the paper.

n/a

Reviewer 3 Report

Comments to the Author(s)

This manuscript presents Experimental Investigation on the Corrosion Detectability of A36 Low Carbon Steel Used in Multi-bolted Connections by the Method of Phased Array Corrosion Mapping. This paper contains a good effort related to NDT systems because these methods can be used to check the integrity of petrochemical plant materials. This paper contains good material of interest to the petrochemical plants and NDT communities. In general, the manuscript is well organized. It has clear and simple text to read. However, the authors are suggested to improve the paper by resolving the following issues (All the answers should be included in the manuscript):

1.      On page 2, line 88, please add a reference to the following paragraph“ (Carbon dioxide generates a type of corrosion…”.

2.      On page 3, line 117, please add a reference to the following paragraph“ (Internal corrosion is caused by gases or liquids that are…”.

3.      On page 4, line 171,  please rewrite the following abbreviated “PAUT” to (phased array ultrasonic testing (PAUT)”.

4.      On page 5, line 231, please separate the length unit and its value by adding a space of the following“The test specimen measured 200mm in length, 100mm in width, and 15mm…”.

5.      Page 6, line 280 please rewrite the caption of Fig. 1” Experiment setup on A36 carbon steel material” to “Schematic of the test specimen design with detailed dimensions”.

6.      In section 2.3. Phased array corrosion mapping, please add the plot of velocity values with a temperature range.

7.      In section 2.3. Phased array corrosion mapping, please add a new figure. In this figure, you will illustrate the schematic of the experimental setup of doing phased array corrosion mapping and label all the exp. items.

8.      Page 10, line 406, please add the figure of the third stage (100°C to 140°C).

9.      Page 11, line 444, please check the number of data sets. I think it should be 69.

10.  I suggest making a separation between the conclusions and the future work.

Minor editing of English language required

Reviewer 4 Report

Please see attached comments.

Reviewer 5 Report

The paper submitted for review is an experimental study of the possibility to detect corrosion damage in pipeline systems using phased array mapping. The problem with the development of corrosion processes in such systems is extremely important from an economic and ecological point of view. Corrosion prevention and monitoring are an integral part of the operation of large facilities.

The work would be of interest not only from a scientific but also from a practical point of view. The methods used are adequately selected for this type of research. The volume of point 1 Introduction is too large considering that the work is not а review. This complicates and confuses the readers, and shifts the focus from the main purpose of the experimental study. I recommend reducing it by addressing the problem specifically. The literature cited is also excessive given the nature of the work. If the composition of the tested steel is determined, it must be presented. It would be good to present the applications of this material in more detail. The quality of the figures presented needs to be checked to see if it meets the journal's requirements. Can figures 3 to 6 be processed rather than taking a direct screenshot of the software? If this is not possible, necessarily improve their resolution. The conclusions should also be reworked, first reducing their volume. They must be precise, clear and reflect the synthesized results.

The English language used in the post needs a little checking to make it sound smoother. Easy to read and understandable.

Round 2

Reviewer 2 Report

The experimental results should redraw by the Matlab, not just capatured from the instrument.

The English is not difficult to understand.
